# Typical pneumonia among human immunodeficiency virus-infected patients in public hospitals in southern Ethiopia

**Ayele Assefa[1], Melat Woldemariam[1]\*, Addis Aklilu[1]\*, Dagninet Alelign[1], Abdurezak Zakir[1], Aseer Manilal[1]\*, Temesgen Mohammed[2], Reham M. Alahmadi[3], Gurusamy Raman[4], Akbar Idhayadhulla[5]**

1 Department of Medical Laboratory Science, College of Medicine and Health Sciences, Arba Minch University, Arba Minch, Ethiopia, 2 School of Public Health, College of Medicine and Health Sciences, Arba Minch University, Arba Minch, Ethiopia, 3 Department of Botany and Microbiology, College of Science, King Saud University, Riyadh, Saudi Arabia, 4 Department of Life Sciences, Yeungnam University, Gyeongsan, Gyeongbuk-Do, South Korea, 5 Research Department of Chemistry, Nehru Memorial College (Affiliated to Bharathidasan University), Puthanampatti, Tiruchirappalli District, Tamil Nadu

\* addaklilu@gmail.com (AA); aseer.manila@amu.edu.et (AM); wolde21mealat@gmail.com (MW)

**Data Availability Statement:** All relevant data are within the manuscript and its Supporting Information files.

## Abstract

### Background

Typical pneumonia is a pressing issue in the treatment of human immunodeficiency virus (HIV) patients, especially in Sub-Saharan Africa, where it remains a significant menace. Addressing this problem is crucial in improving health outcomes and the reduction of the burden of diseases in this vulnerable category of patients.

### Objective

To determine the prevalence of community-acquired typical pneumonia among HIV patients in Public Hospitals in southern Ethiopia.

### Methods

A cross-sectional study was done among 386 HIV patients clinically suspected of typical pneumonia attending the anti-retroviral therapy (ART) clinics of two hospitals from March to September 2022. A pretested structured questionnaire was employed to collect the demographic, clinical, and behavioral data. Sputum samples were collected and inspected for bacteria following standard procedures, and antimicrobial susceptibility testing was performed employing the Kirby-Bauer disk diffusion method. Besides, extended-spectrum β-lactamase (ESβL) and carbapenemase-producing Gram-negative bacteria were inspected by the double disk synergy test and modified carbapenem inactivation method. Descriptive and inferential statistical analyses were also done.

### Results

Overall, 39.1% (151/386) of sputum cultures (95% Confidence Interval: 32.4–44) were bacteriologically positive. A total of 151 bacteria were identified, comprising 72.8% (n = 110)

**Funding:** The author(s) received no specific funding for this work.

**Competing interests:** The authors have declared that no competing interests exist.

of Gram-negative bacteria. The predominant isolate was *Klebsiella pneumoniae* (25.8%, n = 39), followed by *Staphylococcus aureus* (17.9%, n = 27); 59.6% (n = 90) of the entire isolates were multidrug-resistant (MDR). Forty percent (11/27) of *S. aureus* were methicillin-resistant *S. aureus* (MRSA), and 28.1% (n = 31) and 20.9% (n = 23) of Gram-negative bacteria were extended-spectrum beta-lactamases (ESBL) and carbapenemase producers, respectively. Occupational status, alcohol consumption, cluster of differentiation$_4$ (CD$_4$) Thymocyte cell count < 350, interruption of trimethoprim-sulfamethoxazole prophylaxis and antiretroviral treatment, and recent viral load $\geq$ 150 were found statistically significant.

## Conclusion

The higher rates of MDR, MRSA, ESBL, and carbapenem-resistant Enterobacterales (CRE) indicate that bacterial pneumonia is a vexing problem among HIV patients and therefore it is advisable to implement an antimicrobial stewardship program in the study area.

## Introduction

In human immunodeficiency virus (HIV) infected patients, typical or bacterial pneumonia is a common manifestation of immunosuppression [1]. The HIV infection causes alterations in several lines of host defenses in the lungs and the respiratory tract, which contribute to an increased risk of developing typical pneumonia [2]. Anti-retroviral therapy (ART) has revolutionized the management of HIV-associated opportunistic diseases, resulting in significant reductions in their prevalence [3]. However, HIV patients still face a considerable risk of bacterial pneumonia, which is responsible for the majority of hospitalizations [4–6]. A couple of research has shown that the likelihood of experiencing this condition is almost 25 times greater in HIV patients [7,8]. Bacterial pneumonia can be frequently recurrent among HIV patients and is a defining condition of acquired immunodeficiency syndrome (AIDS), as per the Centers for Disease Control and Prevention. It is estimated that a significant proportion of AIDS patients have experienced at least an episode of severe bacterial pneumonia during the course of the disease [9], leading to respiratory failure and other life-threatening complications such as arthritis, hepatitis, meningitis, pericarditis, and empyema if not treated properly [4,9].

It is to be noted that bacterial pneumonia can be an initial indication of underlying HIV infection. Therefore, it is recommended to consider the possibility of latent HIV infection in any individual who presents with recurrent pneumonia, especially when there are no other existing risk factors [10]. In sub-Saharan Africa, bacterial pneumonia remains a critical public health problem, particularly among HIV patients [11], and is associated with an excess of case-fatality rate. It is a fact that *Streptococcus pneumoniae* is one of the most common pathogenic bacteria that can cause pneumonia and is considered to be the most prevalent across all age groups [12]. It is responsible for around 30% of all pneumonia cases. Other types of bacteria that can cause pneumonia include *Klebsiella pneumoniae*, *Haemophilus influenzae*, *Escherichia coli*, *Pseudomonas aeruginosa*, and *Staphylococcus aureus* [4].

Earlier studies have shown that several factors are linked with the incidence of pneumonia among HIV patients. The critical risk factors connected to bacterial pneumonia are recent CD$_4$ T cell count $\leq$350 cells/mm$^3$, alcoholism, HIV-World Health Organization (WHO) Stage II, a viral load $\geq$ 150 copies/mL, smoking, old age, and cooking food using firewood in the

living room [4,11,13]. Therefore, identifying these factors may provide a practical and theoretical basis for the implementation of preventive measures and targeted interventions.

It is worth noting that respiratory tract infections account for the majority of antibiotic prescriptions worldwide. A higher level of antimicrobial resistance is reported in bacteria isolated from HIV patients with pneumonia [13], and hence, they must follow a specific regimen [14]. In Africa, HIV patients often suffer from comorbid illnesses such as respiratory infections, with bacterial pneumonia being the most prevalent and fatal [15]. Even with anti-retroviral and antibiotic treatments, mortality rates associated with bacterial pneumonia in HIV patients can exceed 30% [15]. Drug-resistant bacterial infections can lead to higher rates of morbidity and mortality, which can prolong hospital stays, increase bills, and, in some cases, even result in untreatable infections [16]. The situation is more intricate in several African countries, and the management is extremely challenging due to difficulties in identifying the exact respiratory pathogens and in administering an appropriate regimen [17].

In Ethiopia, opportunistic infections are responsible for over 90% of HIV/AIDS-related deaths, with bacterial pneumonia being the leading cause [18]. Nevertheless, a national data bank on the prevalence of opportunistic infections among HIV/AIDS patients in the country is sorely lacking; studies conducted in selected regions have shown that bacterial pneumonia occurs among HIV patients, and it ranges from 41.7 to 43.7% [4,13]. Typical pneumonia is considered one of the most prevailing opportunistic bacterial infections among HIV patients in the area under study. Unfortunately, current treatment regimens are solely reliant on clinical or radiological diagnosis, which could facilitate the emergence of drug-resistant bacteria. Due to the prolonged use of antibiotics, the emergence and spread of drug-resistant strains will probably make the treatment of pneumonia in HIV patients more complicated. However, there is a shortage of reports on the bacterial profile of typical pneumonia and its antibiogram among HIV patients in the study area. Besides, pertinent dependent factors that influence typical pneumonia have not yet been reported. The present study is executed to address the knowledge gap existing in this context in two government hospitals of Arba Minch, southern Ethiopia. This kind of local/regional study would help generate the latest and most valuable data, which assist in the formulation of intervention protocols.

## Materials and methods

### Study setting, design, period, and population

The study was performed at the ART clinics of Arba Minch General and Dil Fana Primary Hospitals in Arba Minch, southern Ethiopia; the two title hospitals are distinct healthcare facilities that offer ART treatments. The former provides a comprehensive range of ART services, including outpatient and inpatient departments, a pharmacy with a team of experienced ART specialists and supporting staff reputed for giving specialized care. Due to the wide range of services offered, patient flow is consistently on the higher side. Currently, 1,699 adult HIV patients are actively following ART patient care. On the other hand, Dil Fana Primary Hospital offers basic ART services comprising OPD as well as the pharmacy. It has a comparatively smaller team of ART doctors who also have to handle general medical cases, and around 500 adult HIV patients receive ART care. A bi-centric cross-sectional study was performed among all the adult HIV patients (both sexes) clinically diagnosed with typical bacterial pneumonia (community-acquired) seeking medical care from the ART clinics of Arba Minch General and Dil Fana Primary Hospitals from March 1 to September 30, 2022. The criteria of inclusion were 1. all adult HIV patients clinically suspected of typical bacterial pneumonia (i.e., community-acquired pneumonia) (as per the radiography and meticulous clinical examination by ART clinicians); 2. consent to participate in the study. The exclusion criteria were 1. all who

were severely ill and inept to provide sufficient samples, 2. who were diagnosed with pulmonary tuberculosis, 3. who took antibiotics two weeks before the commencement of the study (except for *trimethoprim*-sulfamethoxazole prophylaxis), and 4. those with incomplete medical records.

## Sample size and patient recruitment

We have used a single population proportion formula to calculate the appropriate sample size accurately. A prevalence of 43.7% has been chosen from a prior study conducted in northern Ethiopia [4]. A 95% confidence interval (CI) and 5% margin of error were employed to determine the sample size, which is 378. After incorporating a 10% non-response rate, the total sample size was calculated to be 416. A consecutive sampling method was employed to recruit the eligible participants.

## Data collection

All participants were given a clear explanation of the purpose of the study, and each one of them submitted informed consent prior to data and specimen collection. A structured questionnaire was developed and utilized [4,11,13] through face-to-face interviews to gather information on patient-related socio-demographic characteristics and behavioral factors. The data on clinical and immunological laboratory variables ($CD_4$ T-cell counts, viral load (HIV-1 RNA copies/mL), HIV stage, duration of HIV, etc) were collected after referring the patient's medical records. The medical records of the patients, not more than three months old, were reviewed to obtain the latest $CD_4$ T-cell counts as well as the status of trimethoprim-sulfamethoxazole prophylaxis.

## Specimen collection

Sputum samples (3–5 mL) were collected in a sterile sputum cup by means of standard collection procedures ensuring droplet precautions techniques and were then immediately transported to the Medical Microbiology and Parasitology Laboratory of Arba Minch University. The quality of sputum samples was assessed in terms of Bartlett's scoring [19].

## Bacterial isolation

Sputum samples accepted as per Bartlett's scoring were inoculated onto 5% sheep blood agar, chocolate agar, mannitol salt agar, and MacConkey agar (Oxoid, Basingstoke, Hampshire, UK). The blood and chocolate agar plates were incubated (5–10% $CO_2$) in a candle jar at 37˚ C for 24–48 hours, whereas MacConkey and mannitol salt agar plates were incubated under aerobic conditions at 37˚ C for 24 hours [20]. Isolates were identified phenotypically by examining the colony characteristics (shape, edge, elevation, consistency, density, and color), Gram staining (HiMedia Laboratories Pvt. Ltd, Mumbai, India), motility, and also by means of a series of standard biochemical tests [21]. Briefly mentioning, Gram-positive bacteria were identified using catalase and coagulase tests (*Staphylococcus* sp.). Identification of *Streptococcus pyogenes* was made based on the beta-hemolytic activity on 5% sheep blood agar, catalase test, bacitracin susceptibility (using the 0.04-U disc, Abtek, England), and hydrolysis of pyrrolidonyl aminopeptidase (Hardy Diagnostics, USA) tests. Isolates of *S. pneumoniae* were identified by alpha-hemolytic property, catalase, optochin sensitivity, and bile solubility tests [21].

Isolates of members of the Enterobacteriaceae family *E. coli*, *K. pneumoniae*, *K. oxytoca*, *Enterobacter aerogenes*, and *Proteus* spp., were identified through a series of tests: catalase, indole, citrate, urease, triple-sugar iron, $H_2S$ production, methyl red, and Voges–Proskauer

(HiMedia Laboratories Pvt. Ltd, Mumbai, India). Non–lactose fermenting Gram-negative bacteria (*P. aeruginosa* and *Acinetobacter baumannii*) were identified by indole, citrate, triple-sugar iron, urease, oxidase, and catalase tests [21]. *Haemophilus influenzae* was identified by the formation of colorless-to-grey opaque colonies, which grow in the presence of X and V factors on chocolate agar, showing catalase and oxidase positive, and also by satellite tests [21].

## Antimicrobial susceptibility testing

The Kirby-Bauer disc diffusion technique was employed to inspect the antibiotic susceptibility profile based on the criteria provided by the Clinical and Laboratory Standards Institute (CLSI) [22]. The inocula were prepared by suspending the test organisms in sterile normal saline and adjusting the turbidity to 0.5 McFarland standard; the respective non-fastidious test bacteria were uniformly swabbed over Mueller-Hinton agar plates (Oxoid, Basingstoke, Hampshire, UK). The Mueller-Hinton agar plates enriched with 5% sheep blood were used for fastidious organisms [20,21]. Finally, the discs (Oxoid, Basingstoke, Hampshire, UK) were placed into Petri plates and incubated at 37˚C for 24 hours.

Antibiotic discs for testing were chosen strictly according to the CLSI guidelines and also by considering the prescription policy followed in our study settings. The drug for Gram-positive bacteria was penicillin (PEN) (10μg), amoxicillin-clavulanate (AMC) (20/10μg), cefoxitin (CXT) (30μg), chloramphenicol (CHL) (30μg), ciprofloxacin (CPR) (5μg), clindamycin (CLD) (2μg), gentamicin (GEN) (10μg), trimethoprim-sulfamethoxazole (SXT) (1.25/23.75μg), tetracycline (TET) (30μg) and erythromycin (ERY) (15μg). In the case of Gram-negative bacteria, ampicillin (AMP) (10μg), amoxicillin-clavulanate, piperacillin (PIP) (100 μg), ceftriaxone (CTR) (30μg), cefepime (CFP) (30μg), meropenem (MER) (10μg), gentamicin, chloramphenicol, ciprofloxacin, trimethoprim-sulfamethoxazole, and tetracycline were used. The results were interpreted as susceptible, intermediate, or resistant based on the CLSI guidelines. Multi-drug resistance was defined as the non-susceptibility to at least one agent in $\geq$ 3 classes (categories) of drugs [22].

**Detection of methicillin-resistant *S. aureus*.**   The isolates of *S. aureus* were further subjected to antimicrobial susceptibility tests using cefoxitin discs (surrogate test for oxacillin), and those that showed zones of inhibition <21 mm were considered methicillin-resistant [22].

**Detection of extended-spectrum beta-lactamase (ESBL) producers.**   *Screening and confirmatory tests*. The susceptibility of Gram-negative bacteria was tested against the third generation of cephalosporin, such as ceftriaxone, cefotaxime, and ceftazidime, by using the routine Kirby-Bauer disc diffusion method on Mueller-Hinton agar. Isolates that showed zones of inhibition <25 mm for ceftriaxone (30μg), <27 mm for cefotaxime (30μg), and/or <22 mm for ceftazidime (30μg) were considered positive for ESBL production [22].

A disc of amoxicillin-clavulanate (30μg) with third-generation cephalosporins such as ceftriaxone (30μg), cefotaxime (30μg), and ceftazidime (30μg) were used for confirmation. The amoxicillin-clavulanate disc was placed at the center of the inoculum on Mueller-Hinton agar media, and discs of ceftriaxone, cefotaxime, and ceftazidime were placed 15 mm away from the central one. An enhanced zone of inhibition produced towards the amoxicillin-clavulanate disc is the characteristic of a potential ESBL producer [22].

**Phenotypic identification of carbapenemase producers.**   Mentioning briefly, the test isolates of respective species of Gram-negative bacteria (1 μL loop-full of *Enterobacterales* or 10 μL loop of *P. aeruginosa*) were suspended in 2 mL of trypticase soy broth, and a meropenem disc was aseptically placed and incubated at 37˚C for 4 hours in ambient air. A suspension of meropenem susceptible indicator strain of *E. coli* (ATCC 25922) equivalent to 0.5 McFarland standard was swabbed onto Mueller Hinton agar. The incubated disc was then transferred

from the broth suspension to the Mueller Hinton agar and re-incubated at 37˚C for 24 hours. If the isolate produces carbapenemase, the meropenem in the disc undergoes hydrolysis, and there is either no zone of inhibition or only a limited enhancement in the inhibition zone corresponding to the meropenem susceptible *E. coli* (ATCC 25922). Results were classified as positive for the inhibition zone with a diameter of 6–15 mm, intermediate for 16–18 mm, and negative for ≥19 mm [22].

## Data quality assurance

The structured questionnaire was pre-tested on 5% of the sample size at the Chencha General Hospital. The data collectors and technicians were given sufficient training sessions before collecting the actual data and samples, as per standard procedures. The data were checked daily for accuracy, clarity, and completeness, and any incompleteness and errors found were immediately corrected with utmost confidence. Standard operating procedures (in-house SOP manual) were followed during the collection, transportation, and processing to maintain the highest level of quality. All reagents, culture media, and antibiotic discs were carefully inspected for their shelf life and physical condition and were stored at 2–8˚C. The culture media were incubated at 37˚C overnight to ensure sterility until actual sample processing. The efficacy of the media was determined by inoculating the American Type Culture Collection (ATCC) ((*S. aureus* (ATCC 25923) *E. coli* (ATCC 25922), and *P. aeruginosa* (ATCC 27853)).

## Statistical analysis

The IBM SPSS Statistics for Windows, version 25 (IBM Corp., Armonk, N.Y., USA) was used for data analysis. The dependent variable in the analysis was culture-proven bacterial pneumonia. The socio-demographic and clinical factors were initially described using descriptive statistics. Inferential statistical analysis was conducted, which included bivariable and multivariable binary logistic regression. The data were subjected to bivariable analyses initially, and only those covariates with a p-value ≤ 0.25 were further processed through multivariable analysis; the Hosmer-Lemeshow goodness fit test was employed to assess the fitness of the model. The strength of association was determined using adjusted odds ratio (AOR) and 95% confidence interval (CI), and statistical significance was assigned only to p-values ≤ 0.05.

## Ethical considerations

The study protocol (Ref No. IRB/1201/2022) was approved by the Institutional Research Ethics Review Board of the College of Medicine and Health Science of Arba Minch University. Each participant submitted written informed consent before the actual collection of data and samples, as mentioned earlier. The results of the positive sputum culture and antimicrobial susceptibility patterns were promptly communicated to the ART physicians to arrange the proper orientations and fix treatments as well as follow-up. All necessary measures were taken throughout to ensure the ethical conduct of the study.

## Results

### Socio-demographic and clinical factors

A total of 416 cases of typical pneumonia were enrolled in this study; 386 fulfilled the eligibility criteria as per the sputum quality (Bartlett's score), and 75.1% (n = 290) of the participants were urban dwellers; mean age was 39.3±9.9, and more than half of them were females, 52.3% (n = 202); majority of participants, 68.1% (n = 263), were married. The majority, i.e., 61.9% (n = 239) of them, had BMI in the range of 18.5–24.9 kg/m$^2$, and 36.8% (n = 142) had a recent

**Table 1. Magnitude of typical pneumonia among HIV patients.**

| Variables | Categories | Occurrence of bacterial pneumonia n (%) | | |
|---|---|---|---|---|
| | | Total | Positive | Negative |
| Sex | Male | 184 (47.7) | 76 (41.3) | 108 (58.7) |
| | Female | 202 (52.3) | 75 (37.1) | 127 (62.9) |
| Age [years] | 18–29 | 61 (15.8) | 19 (31.1) | 42 (68.9) |
| | 30–39 | 141 (36.5) | 48 (34) | 93 (66) |
| | 40–49 | 106 (27.5) | 50 (47.2) | 56 (52.8) |
| | ≥50 | 78 (20.2) | 34 (43.6) | 44 (56.4) |
| Residence | Urban | 290 (75.1) | 110 (37.9) | 180 (62.1) |
| | Rural | 96 (24.9) | 41 (42.7) | 55 (57.3) |
| Marital status | Single | 50 (13) | 18 (36) | 32 (64) |
| | Married | 263 (68.1) | 101 (38.4) | 162 (61.6) |
| | Divorced | 43 (11.1) | 18 (41.9) | 25 (58.1) |
| | Widowed | 30 (7.8) | 14 (46.7) | 16 (53.3) |
| Educational status | Illiterate | 40 (10.4) | 20 (50) | 20 (50) |
| | Elementary school | 75 (19.4) | 31 (41.3) | 44 (58.7) |
| | High school | 191 (49.5) | 69 (36.1) | 122 (63.9) |
| | Diploma & above | 80 (20.7) | 31 (38.8) | 49 (61.3) |
| Occupational status | Working actively | 314 (81.3) | 116 (36.9) | 198 (63.1) |
| | Unemployed | 72 (18.7) | 35 (48.6) | 37 (51.4) |
| Body mass index (kg/m$^2$) | <18.5 | 56 (14.5) | 29 (51.8) | 27 (48.2) |
| | 18.5–24.9 | 239 (61.9) | 89 (37.2) | 150 (62.8) |
| | ≥25.0 | 91 (23.6) | 33 (36.3) | 58 (63.7) |
| A habit of mouth washing and brushing | Never | 130 (33.7) | 69 (53.1) | 61 (46.9) |
| | Sometimes | 202 (52.3) | 64 (31.9) | 138 (68.3) |
| | Always | 54 (14) | 18 (33.3) | 36 (66.7) |
| Alcohol consumption | Yes | 126 (32.6) | 73 (57.9) | 53 (42.1) |
| | No | 260 (53.4) | 78 (30) | 182 (70) |
| Smoking | Yes | 46 (11.9) | 33 (71.7) | 13 (28.3) |
| | No | 340 (88.1) | 118 (34.7) | 222 (65.3) |
| Cotrimoxazole prophylaxis use | Yes | 297 (76.9) | 90 (30.3) | 207 (69.7) |
| | No | 89 (23.1) | 61 (68.5) | 28 (31.5) |
| Recent CD$_4$ T count (cells/mm$^3$) | >500 | 105 (27.2) | 22 (21) | 83 (79) |
| | 350–500 | 142 (36.8) | 45 (31.7) | 97 (68.3) |
| | <350 | 139 (36) | 84 (60.4) | 55 (39.6) |
| Viral load (copies/mL) | Undetectable | 254 (65.8) | 72 (28.3) | 182 (71.7) |
| | <150 | 106 (27.5) | 60 (56.6) | 46 (43.4) |
| | ≥150 | 26 (6.7) | 19 (73.1) | 7 (26.9) |
| WHO HIV stages | I | 219 (56.7) | 57 (26) | 162 (74) |
| | II | 115 (30) | 61 (53) | 54 (47) |
| | III and IV | 52 (13.5) | 33 (63.5) | 19 (36.5) |

CD$_4$ T cell count of 350–500 cells/mm$^3$. Most of the participants (56.7%, n = 219) are in WHO clinical stage I of HIV (Table 1).

## Prevalence of typical pneumonia

The overall prevalence of community-acquired typical pneumonia was 39.1% (95% CI: 32.4–44). Typical pneumonia can impact all age groups of patients with immunosuppression. In our study, it was prevalent among participants aged 40–49 (n = 50, 47.2%) and rare among the age group 18–29 (n = 19, 31.1%). Pneumonia was more prevalent among male participants, i.e., 41.3% (n = 76). Among the 126 participants who had the habit of consuming alcohol, 57.9% (n = 73) were culture-positive for pneumonia. Also, 37.9% (n = 110) of urban residents were culture-positive (Table 1).

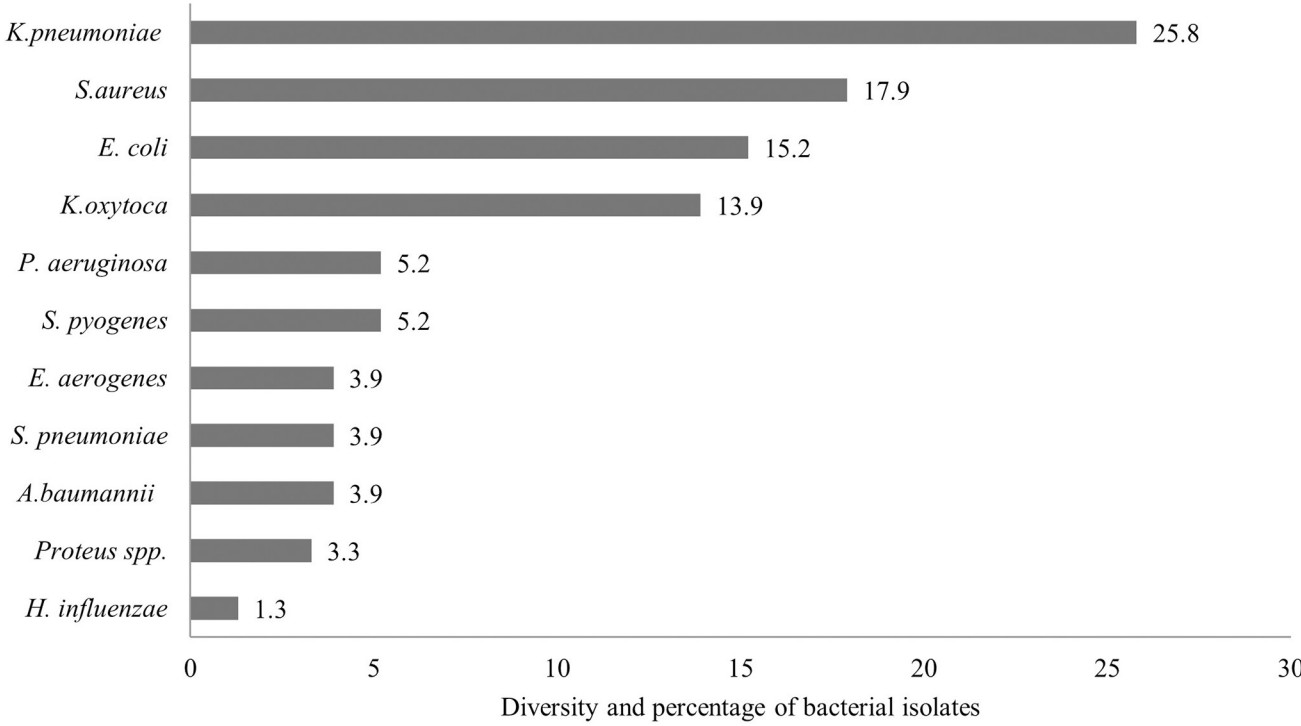

**Fig 1. Bacterial profile of sputum culture.**

## Bacteriological profile

With respect to the types of bacteria associated with typical pneumonia, 151 isolates were detected; the most common were Gram-negative bacteria, 72.8% (110/151), while Gram-positive counterparts accounted for only 27.1% (41/151). The most common causative agent was *K. pneumoniae* (25.8%), followed by *S. aureus* (17.9%), *E. coli* (15.2%), *S. pyogenes* (5.2%), *P. aeruginosa* (5.2%), *E. aerogenes* (3.9%), and *S. pneumoniae* (3.9%) (Fig 1). Notably, only one sputum sample had a bi-bacterial infection, and the pair involved were *K. pneumoniae* and *E. coli*.

## Antimicrobial susceptibility profiles of Gram-positive bacteria

Gram-positive bacteria exhibited resistance to tetracycline, 43.9% (18/41), erythromycin, 46.3% (19/41), penicillin, 60% (21/35), and trimethoprim-sulfamethoxazole, 39% (16/41). It was observed that higher percentages of susceptibility were shown against various antibiotics, such as chloramphenicol (87.8%) and clindamycin (73.2%). Higher percentages of resistance were exhibited by *S. aureus* isolates against penicillin (77.8%), erythromycin (55.6%), trimethoprim-sulfamethoxazole (37%), and cefoxitin (40.7%). However, chloramphenicol (88.9%) and gentamicin (74.1%) were found to be highly effective. The percentage of MRSA was 40.7 (11/27). Isolates of *S. pneumoniae* demonstrated only moderate resistance to erythromycin and tetracycline, i.e., 33.3% each (2/6), and relatively higher extent of resistance to trimethoprim-sulfamethoxazole, 66.7% (4/6). Interestingly, they were more susceptible to chloramphenicol, amoxicillin-clavulanate, and clindamycin, i.e., 83.3% each (5/6). The isolates of *S. pyogenes* showed a higher resistance level to tetracycline, 87.5% (7/8), whereas the susceptibility to chloramphenicol and clindamycin was 87.5% each (7/8), and in the case of erythromycin, it was 75% (6/8) (Table 2).

**Table 2. Antibiotic susceptibility profiles of Gram-positive bacteria.**

| Bacterial isolates | Susceptibility patterns | Antimicrobial agents n (%) | | | | | | | | | |
|---|---|---|---|---|---|---|---|---|---|---|---|
| | | AMC | CPR | CHL | CXT | CLD | ERY | PEN | TET | SXT | GEN |
| *S. aureus* (n = 27) | S | NT | 18(66.7) | 24(88.9) | 16(59.3) | 18(66.7) | 12(44.4) | 6(22.2) | 18(66.7) | 17(63) | 20(74.1) |
| | R | NT | 9(33.3) | 3(11.1) | 11(40.7) | 9(33.3) | 15(55.6) | 21(77.8) | 9(33.3) | 10(37) | 7(25.9) |
| *S. pneumoniae* (n = 6) | S | 5(83.3) | NT | 5(83.3) | NT | 5(83.3) | 4 (66.7) | NT | 4 (66.7) | 2 (33.3) | NT |
| | R | 1(16.7) | NT | 1(16.7) | NT | 1(16.7) | 2 (33.3) | NT | 2 (33.3) | 4 (66.7) | NT |
| *S. pyogenes* (n = 8) | S | NT | NT | 7(87.5) | NT | 7(87.5) | 6(75) | 8(100) | 1 (12.5) | 6 (75) | NT |
| | R | NT | NT | 1 (12.5) | NT | 1(12.5) | 2(25) | - | 7(87.5) | 2 (25) | NT |
| **Total tested** | | 6 | 27 | 41 | 27 | 41 | 41 | 35 | 41 | 41 | 27 |
| **Total Susceptible** | | 5(83.3) | 18(66.7) | 36(87.8) | 16(59.3) | 30(73.2) | 22(53.7) | 14(40) | 23(56.1) | 25(61) | 20(74.1) |
| **Total Resistant** | | 1(16.7) | 9(33.3) | 5(12.2) | 11(40.7) | 11(26.8) | 19(46.3) | 21(60) | 18(43.9) | 16(39) | 7(25.9) |

NT: Not tested, AMC: Amoxicillin-clavulanate, CPR: Ciprofloxacin, CHL: Chloramphenicol, CXT: Cefoxitin, CLD: Clindamycin, ERY: Erythromycin, PEN: Penicillin, TET: Tetracycline, SXT: Trimethoprim-sulfamethoxazole, GEN: Gentamicin.

## Antimicrobial susceptibility profiles of Gram-negative bacteria

Gram-negative bacteria exhibited a broader range of antimicrobial resistance, spanning 26.9 to 71.6%. They showed resistance to trimethoprim-sulfamethoxazole, 58.2% (64/110), tetracycline, 71.6% (73/102), amoxicillin-clavulanate, 64.3% (63/98), and meropenem, 50% (55/110), however were moderately susceptible to chloramphenicol, 57.3% (55/96), piperacillin, 73.1% (79/108), ampicillin, 56% (14/25), and cefepime, 62.7% (69/110).

The isolates of *K. pneumoniae* were resistant to ciprofloxacin, 87.2% (34/39), tetracycline, 82% (32/39), gentamicin, 64.1% (25/39), amoxicillin-clavulanate, trimethoprim-sulfamethoxazole, each 71.8% (28/39), and meropenem, 61.5% (24/39) (Table 3). Conversely, piperacillin and cefepime were the most active antibiotics against *K. pneumoniae*, 87.2% (34/39) and 74.4% (29/39), respectively. On the other hand, isolates of *P. aeruginosa* exhibited resistance to five antibiotics tested, and the extent of resistance ranged from 12.5 to 87.5%. Tetracycline resistance was observed in 87% (20/23) of *E. coli* isolates, while 65.2% (15/23) of them were resistant to trimethoprim-sulfamethoxazole. Meanwhile, a moderate level of resistance was exhibited against meropenem and amoxicillin-clavulanate, 56.5% each (13/23). It is to be noted that *E. coli* isolates had higher susceptibility to piperacillin, 82.6% (19/23), cefepime, 73.9% (17/23), and chloramphenicol, 87% (20/23). The *E. aerogenes* isolates showed equal resistance, i.e., 66.7% (4/6), to trimethoprim-sulfamethoxazole and meropenem. In addition, these bacteria exhibited gentamicin resistance at a medium level (i.e., 50%, 3/6). Nevertheless, chloramphenicol and tetracycline were the most active antibiotics against them, with a susceptibility of 83.3% (5/6) for each.

## Multi-drug resistance patterns

This study classified the MDR as the resistance to three or more groups of antibiotics tested, and accordingly, 59.6% (n = 90) of bacteria were MDR; of these, 74.4% (67/90) were Gram-negative bacteria, the most common MDR isolate was *K. pneumoniae*, 59% (23/39). All the isolates of *A. baumannii*, 100% (6/6), and *H. influenzae*, 100% (2/2), were MDR. The Gram-positive MDR bacteria comprise 50% (3/6) of *S. pneumoniae*, 15.9% (14/27) of *S. aureus*, and 4.5% (4/8) of *S. pyogenes* isolates (Table 4).

**Table 3. Antibiotic susceptibility profiles of Gram-negative bacteria.**

| Bacterial isolates | Susceptibility pattern | Antimicrobial agents n (%) | | | | | | | | | | |
|---|---|---|---|---|---|---|---|---|---|---|---|---|
| | | AMC | AMP | PIP | CTR | CFP | GEN | CHL | CPR | MER | TET | SXT |
| E. coli (n = 23) | S | 10(43.5) | 13(56.5) | 19(82.6) | 13(56.5) | 17(73.9) | 16(69.6) | 20(87) | 15(65.2) | 10(43.5) | 3(13) | 8(34.8) |
| | R | 13(56.5) | 10(43.5) | 4(17.4) | 10(43.5) | 6(26.1) | 7(30.4) | 3(13) | 8(34.8) | 13(56.5) | 20(87) | 15(65.2) |
| K. pneumoniae (n = 39) | S | 11(28.2) | NT | 34(87.2) | 26(66.7) | 29(74.4) | 14(35.9) | 14(35.9) | 5(12.5) | 15(38.5) | 7(18) | 11(28.2) |
| | R | 28(71.8) | NT | 5(12.8) | 13(33.3) | 10(25.6) | 25(64.1) | 25(64.1) | 34(87.2) | 24(61.5) | 32(82) | 28(71.8) |
| P. aeruginosa (n = 8) | S | 1(12.5) | NT | 3(37.5) | 5(62.5) | 6(75) | 2(25) | NT | 4(50) | 7(87.5) | NT | 3(37.5) |
| | R | 7(87.5) | NT | 5(62.5) | 3(37.5) | 2(25) | 6(75) | NT | 4(50) | 1(12.5) | NT | 5(62.5) |
| Proteus spp. (n = 5) | S | 3(60) | NT | 3(60) | 3(60) | 4(80) | 2(40) | 3(60) | 3(60) | 2(40) | 3(60) | 2(40) |
| | R | 2(40) | NT | 2(40) | 2(40) | 1(20) | 3(60) | 2(40) | 2(40) | 3(60) | 2(40) | 3(60) |
| A.baumannii (n = 6) | S | NT | NT | 2(33.3) | 2(33.3) | 3(50) | 3(50) | NT | 2(33.3) | 2(33.3) | 4(66.7) | 2(33.3) |
| | R | NT | NT | 4(66.7) | 4(66.7) | 3(50) | 3(50) | NT | 4(66.7) | 4(66.7) | 2(33.3) | 4(66.7) |
| E. aerogenes (n = 6) | S | NT | NT | 4(66.7) | 5(83.3) | 4(66.7) | 3(50) | 5(83.3) | 4(66.7) | 2(33.3) | 5(83.3) | 2(33.3) |
| | R | NT | NT | 2(33.3) | 1(16.7) | 2(33.3) | 3(50) | 1(16.7) | 2(33.3) | 4(66.7) | 1(16.7) | 4(66.7) |
| H.influenzae (n = 2) | S | 2(100) | 1(50) | NT | 2(100) | 2(100) | 1(50) | 2(100) | 2(100) | 1(50) | 1(50) | 1(50) |
| | R | 0 | 1(50) | NT | 0 | 0 | 1(50) | 0 | 0 | 1(50) | 1(50) | 1(50) |
| K. oxytoca (n = 21) | S | 8(38.1) | NT | 14(66.7) | 3(14.3) | 4(19) | 6(28.6) | 11(52.4) | 7(33.3) | 16(76.2) | 6(28.6) | 17(81) |
| | R | 13(61.9) | NT | 7(33.3) | 18(85.7) | 17(81) | 15(71.4) | 10(47.6) | 14(66.7) | 5(23.8) | 15(71.4) | 4(19) |
| **Total tested** | | 98 | 25 | 108 | 110 | 110 | 110 | 96 | 110 | 110 | 102 | 110 |
| **Total susceptible** | | 35(35.7) | 14(56) | 79(73.1) | 59(53.6) | 69(62.7) | 47(42.7) | 55(57.3) | 43(39.1) | 55(50) | 29(28.4) | 46(41.8) |
| **Total resistant** | | 63(64.3) | 11(44) | 29(26.9) | 51(46.4) | 41(37.3) | 63(57.3) | 41(42.7) | 67(60.9) | 55(50) | 73(71.6) | 64(58.2) |

NT: Not tested, AMC: Amoxicillin-clavulanate, AMP: Ampicillin, PIP: Piperacillin, CTR: Ceftriaxone, CFP: Cefepime, GEN: Gentamicin, CHL: Chloramphenicol, CPR: Ciprofloxacin, MER: Meropenem, TET: Tetracycline, SXT: Trimethoprim-sulfamethoxazole.

## ESBLs and carbapenemase producers

Of the total 110 Gram-negative bacteria tested, 28.1% (31/110) were found to be ESBL producers and carbapenemase production was detected in the case of 20.9% (23/110). Meanwhile, 5.4% (6/110) of the isolates were co-producers of ESBL and carbapenemase; ESBL production

**Table 4. ESBLs and carbapenemase enzyme production patterns of bacteria.**

| Bacterial isolates | MDR n (%) | ESBL producers n (%) | Carbapenemase producers n (%) | ESBL and carbapenemase co-producers n (%) |
|---|---|---|---|---|
| K. pneumoniae (n = 39) | 23 (59) | 8 (20.5) | 6 (15.4) | 2 (5.1) |
| K. oxytoca (n = 21) | 12(57.1) | 9 (42.8) | 4 (19) | 1 (4.7) |
| E. coli (n = 23) | 15(65.2) | 4 (17.4) | 3 (13) | 3 (13) |
| Proteus spp. (n = 5) | 2 (40) | 3(60) | 1(20) | 0(0) |
| A.baumannii (n = 6) | 6(100) | 3 (50) | 2 (33.3) | 0 (0) |
| H. influenzae (n = 2) | 2(100) | - | - | - |
| P. aeruginosa (n = 8) | 5(62.5) | 2 (25) | 5 (62.5) | 0 (0) |
| E. aerogenes (n = 6) | 4(66.7) | 2 (33.3) | 2 (33.3) | 0 (0) |
| S. aureus (27) | 14(51.9) | NT | NT | NT |
| S. pneumoniae (6) | 3(50) | NT | NT | NT |
| S. pyogenes (8) | 4(50) | NT | NT | NT |
| Total (n = 151) | 90(59.6) | 31(28.1) | 23(20.9) | 6 (5.4) |

MDR: Multidrug resistance, ESBL: Extended-spectrum β-lactamase, NT: Not tested.

was confirmed in 20.5% (8/39) of *K. pneumoniae*, 17.4% (4/23) of *E. coli*, 42.8% (9/21) of *K. oxytoca* and 60% (3/5) of *Proteus* spp. Carbapenemase production was found in 13% (3/23) of *E. coli*, 20% (1/5) of *Proteus* spp, and 15.4% (6/39) of *K. pneumoniae*. The co-production of ESBL and carbapenemase was observed in 5.1% (n = 2) and 13% (n = 3) of *K. pneumoniae* and *E. coli*, respectively (Table 4). These findings highlight the urgent need for an increased awareness and action plan to combat the spread of antibiotic resistance.

## Factors associated with typical pneumonia

Age, occupation, trimethoprim-sulfamethoxazole prophylaxis, history of pneumonia, history of tuberculosis, body mass index, habit of mouth washing and brushing, recent $CD_4$ T cell counts and viral load, smoking habits, WHO HIV stage, alcohol consumption, and interruption of ART were significantly associated with bacterial pneumonia according to the bivariable logistic regression analysis done and hence were further subjected to multivariable logistic regression analysis, as candidate variables to control the confounding effect (Table 5).

In the multivariable logistic regression analysis, occupational status (p = 0.02), recent viral load $\geq$ 150 copies/mL (p = 0.002), absence of trimethoprim-sulfamethoxazole as prophylaxis (p$\leq$0.001), recent $CD_4$ T cell count $\leq$350 (p = 0.001), alcohol consumption (p = 0.001), and interruption of ART (p = 0.04) were statistically associated with typical pneumonia. The covariates, such as smoking habit, previous episodes of tuberculosis and pneumonia, and BMI, were not associated with the likelihood of bacterial pneumonia in this analysis.

## Discussion

Our findings indicate that, despite the availability of potent antiretroviral therapy, bacterial pneumonia remains a significant contributor to morbidity in patients who have HIV, suggesting that further efforts are required to combat the challenge. A timely diagnosis and prompt treatment of bacterial pneumonia may hasten the clinical recovery, alleviating the risk of disease progression. There has been a significant increase in antibiotic resistance among pathogens isolated from HIV patients with bacterial pneumonia in recent years, highlighting the urgent need for effective measures to combat the severity [23]. This report presents the first accounts of the prevalence and bacterial profile of typical pneumonia among HIV patients in Arba Minch. The HIV patients in our study area experienced high rates of typical pneumonia, with an overall prevalence of 39.1% (95% CI: 32.4–44), and is consistent with a couple of earlier studies done in the country (Bahirdar (41.7%) and Mekele (43.7%)) [4,13]. A higher prevalence rate might be ascribed to reduced protective immunity, enhancing the vulnerability to respiratory infections [11]. The results obtained are marginally higher than those done in Nepal (24 and 25.5%) [24,25], Nigeria (20.2%) [26], and also from an earlier study done in Ethiopia itself (21.5%) [27], however, some other works conducted in Dessie in northeast Ethiopia (46.3%) [28], Nigeria (55.6%) [29], South Africa (60%) [30], and India (45.1% and 82.9%) [31,32], hint at higher prevalence rate. The fluctuations in prevalence rates can be ascribed to several factors, such as mismatch in sample sizes, socio-demographic features of participants, type of specimens used for diagnosis, and the immune status of individuals [4]. The culture results can become negative because of the fastidious nature of organisms, loss of viability during transport, previous exposure to antibiotics, hampering of growth by normal flora, and a prolonged incubation period.

The prevalence of respiratory tract infections rises over a specific period owing to the increased immunosuppressive conditions following HIV infection. The current study reveals a shift in the types of bacteria, i.e., from mostly Gram-positive to Gram-negative bacteria, the

**Table 5. Bivariable and multivariable logistic regression analyses of factors associated with HIV patients with typical pneumonia.**

| Variable | Category | Sputum culture status | | COR (95% CI) | P-value | AOR (95% CI) | P-value |
|---|---|---|---|---|---|---|---|
| | | Positive n (%) | Negative n (%) | | | | |
| Age (in years) | 18–29 | 19 (31.1) | 42 (68.9) | 1 | | 1 | |
| | 30–39 | 48 (34) | 93 (66) | 0.87 (0.46, 1.67) | 0.69 | 0.80 (0.35, 1.84) | 0.60 |
| | 40–49 | 50 (47.2) | 56 (52.8) | 0.50 (0.26, 0.98) | 0.044* | 0.49 (0.20, 1.18) | 0.11 |
| | ≥50 | 34 (43.6) | 44 (56.4) | 0.58 (0.29, 1.18) | 0.14* | 0.61 (0.24, 0.56) | 0.30 |
| Sex | Male | 76 (41.3) | 108 (58.7) | 0.83 (0.55, 1.26) | 0.40 | - | - |
| | Female | 75 (37.1) | 127 (62.9) | 1 | | | |
| Residence | Urban | 110 (37.9) | 180 (62.1) | 1 | | - | - |
| | Rural | 41 (42.7) | 55 (57.3) | 0.82 (0.51, 1.31) | 0.41 | | |
| Marital status | Single | 18 (36) | 32 (64) | 1 | | | |
| | Married | 101 (38.4) | 162 (61.6) | 0.90 (0.48, 1.69) | 0.75 | - | - |
| | Divorced | 18 (41.9) | 25 (58.1) | 0.78 (0.33, 1.80) | 0.56 | | |
| | Widowed | 14 (46.7) | 16 (53.3) | 0.64 (0.25, 1.61) | 0.34 | | |
| Educational status | Illiterate | 20 (50) | 20 (50) | 0.63 (0.29, 1.36) | 0.24 | 0.55 (0.18, 1.61) | 0.27 |
| | Primary education | 31 (41.3) | 44 (58.7) | 0.89 (0.47, 1.71) | 0.74 | 1.08 (0.44, 2.62) | 0.85 |
| | Secondary education | 69 (36.1) | 122 (63.9) | 1.12 (0.65, 1.92) | 0.68 | 1.46 (0.70, 3.01) | 0.30 |
| | Diploma and above | 31 (38.8) | 49 (61.3) | 1 | | 1 | |
| Occupational status | Employed | 116 (36.9) | 198 (63.1) | 1 | | 1 | |
| | Unemployed | 35 (48.6) | 37 (51.4) | 0.62 (0.37, 1.03) | 0.069* | 0.46 (0.23, 0.93) | 0.029** |
| Cotrimoxazole prophylaxis | Yes | 90 (30.3) | 207 (69.7) | 1 | | 1 | |
| | No | 61 (68.5) | 28 (31.5) | 0.20 (0.12, 0.33) | <0.001* | 0.21 (0.11, 0.40) | 0.001** |
| Previous pneumonia | Yes | 36 (46.8) | 41 (53.2) | 0.67 (0.41, 1.12) | 0.13* | 0.91 (0.46, 1.78) | 0.78 |
| | No | 115 (37.2) | 194 (62.8) | 1 | | 1 | |
| History of tuberculosis | Yes | 12 (54.5) | 10 (45.5) | 0.52 (0.22, 1.22) | 0.13* | 0.58 (0.18, 1.84) | 0.35 |
| | No | 139 (38.2) | 225 (61.8) | 1 | | 1 | |
| BMI | <18.5 | 29 (51.8) | 27 (48.2) | 0.53 (0.26, 1.04) | 0.06* | 0.85 (0.35, 2.06) | 0.72 |
| | 18.5–24.9 | 89 (37.2) | 150 (62.8) | 0.96 (0.58, 1.58) | 0.87 | 0.91 (0.47, 1.73) | 0.77 |
| | ≥25.0 | 33 (36.3) | 58 (63.7) | 1 | | 1 | |
| A habit of mouth washing and brushing | Never | 69 (53.1) | 61 (46.9) | 0.44 (0.23, 0.85) | 0.016* | 0.55 (0.23, 1.30) | 0.17 |
| | Sometimes | 64 (31.7) | 138 (68.3) | 1.07 (0.56, 2.04) | 0.82 | 0.83 (0.36, 1.91) | 0.67 |
| | Always | 18 (33.3) | 36 (66.7) | 1 | | 1 | |
| Recent CD$_4$ T cell count | >500 | 22 (21) | 83 (79) | 1 | | 1 | |
| | 350–500 | 45 (31.7) | 97 (68.3) | 0.57 (0.32, 1.03) | 0.062* | 0.53 (0.25, 1.09) | 0.08 |
| | <350 | 84 (60.4) | 55 (39.6) | 0.17 (0.09, 0.31) | <0.001* | 0.23 (0.11, 0.49) | <0.001** |
| WHO-HIV stages | Stage I | 57 (26) | 162 (74) | 1 | | 1 | |
| | Stage II | 61 (53) | 54 (47) | 0.32 (0.19, 0.51) | <0.001* | 0.55 (0.29, 1.03) | 0.063 |
| | Stage III and IV | 33 (63.5) | 19 (36.5) | 0.21 (0.11, 0.40) | <0.001* | 0.44 (0.20, 1.004) | 0.051 |
| Recent viral load | Undetectable | 72 (28.3) | 182 (71.7) | 1 | | 1 | |
| | <150 | 60 (56.6) | 46 (43.4) | 0.30 (0.18, 0.48) | <0.001* | 0.53 (0.29, 0.95) | 0.035 |
| | ≥150 | 19 (73.1) | 7 (26.9) | 0.15 (0.05, 0.36) | <0.001* | 0.17 (0.05, 0.52) | 0.002** |
| Smoking status | Yes | 33 (21.9) | 13 (5.5) | 0.21 (0.11, 0.41) | <0.001* | 0.44 (0.19, 1.04) | 0.062 |
| | No | 118 (78.1) | 222 (94.5) | 1 | | 1 | |

(*Continued*)

**Table 5.** (Continued)

| Variable | Category | Sputum culture status | | COR (95% CI) | P-value | AOR (95% CI) | P-value |
|---|---|---|---|---|---|---|---|
| | | Positive n (%) | Negative n (%) | | | | |
| Alcohol consumption | Yes | 73 (48.3) | 53 (22.6) | 0.31 (0.20, 48) | <0.001* | 0.38 (0.21, 0.69) | 0.001** |
| | No | 78 (51.7) | 182 (77.4) | 1 | | 1 | |
| Khat chewing status | Yes | 22 (14.6) | 31 (13.2) | 0.89 (0.49, 1.61) | 0.70 | - | - |
| | No | 129 (85.4) | 204 (86.8) | 1 | | | |
| Ever interrupted ART | Yes | 46 (30.5) | 18 (7.7) | 0.18 (0.10, 0.34) | <0.001* | 0.46 (0.22, 0.96) 1 | 0.04** |
| | No | 105 (69.5) | 217 (92.3) | 1 | | | |

*Statistically significant at P≤0.25

** statistically significant at P≤0.05, AOR: Adjusted odds ratio, COR: Crude odds ratio, 1: Reference group, CI: Confidence interval.

latter amounting to 72.8% (110/151). This trend is at par with the outcome of many studies done in Ethiopia itself [4,13,28], Nigeria [29], and Nepal [25].

The lack of awareness of healthcare workers as well as patients, on droplet precautions to be taken and the high environmental burden caused by Gram-negative bacteria, could be the factors contributing to the paradigm shift from Gram positives to Gram negatives. In fact, this shift observed in the results can provide an indication in fixing the appropriate antibiotic regimens. Further in-depth studies involving different settings and larger size study populations are required to arrive at a firm conclusion regarding the paradigm shift at a national level.

The most common bacteria isolated in the current study was *K. pneumoniae* (n = 39), as in the case of a couple of studies done earlier in Ethiopia [4,13] and a study from Nepal [25], whereas it is in contrast with a previous report from Dessie, where *S. pneumoniae* (26.3%) was identified as the predominant one [28]. It is known that *K. pneumoniae* is a common inhabitant of oropharynx and if aspirated, it has the potential to cause severe damage to lungs and can pose greater risks to those with weakened immunity. The observed predominance of *K. pneumoniae* could be linked to its higher degree of infectivity and pathogenicity along with their enhanced drug resistance [16].

The second predominant causative agent in this study was *S. aureus*, which accounted for 17.9%. Again, this corresponds to a previous study undertaken in the country [13] and also in South Africa [30]. This could be partly due to their widespread distribution in hospitals, where they can be spread to the community through contaminated hands of medical staff and patients in addition to being prevailing in the environment and on the mucous membranes of patients [33]. In our study, as in previously published data [4,25], *E. coli* was the second most dominant Gram-negative bacteria isolated, accounting for 15.2%. The existence of *E. coli* in the lower respiratory tract could be due to the micro-aspiration of formerly colonized upper airway secretions in the case of critically ill patients. Besides, microorganisms can also enter the lungs via contaminated food items [29]. An earlier study done in the country among HIV patients found that *K. pneumoniae* (47.4%) was the second most common pathogen [28].

The second most frequently isolated Gram-positive bacteria was *S. pyogenes* (5.2%), and this is comparable to the outcome of a prior study done in India [32]. In this study, *S. pneumoniae* (3.9%), *H. influenzae* (1.3%), and *Proteus* spp. (3.3%) were the rare cause of typical pneumonia, contrary to what was reported in a previous Ethiopian study [28]. This finding also aligns with the conclusions of two studies done in Ethiopia, such as one from Mekele [4] and another from Bahirdar [13]. However, no data on pneumococcal and Hib vaccination among participants were available. Nonetheless, these two causative respiratory pathogens were the

least documented cases of bacterial pneumonia in the study settings. A previous work done among the same study population reported that the extent of nasopharyngeal colonization by *S. pneumoniae* was 13.5% [34]. Several factors could contribute to the occurrence of these pathogens among HIV patients, comprising bacteria acquired from an already contaminated environment, hospitalization before the onset of illness, recurrent bacterial pneumonia, contamination from food consumed, and the systemic spread from blood streams to the lungs [35].

Antibiotics have a salutary effect in treating and preventing opportunistic and associated infections in HIV patients; however, excessive administration, as well as cross-resistance existing between antibiotics, may contribute to the emergence of drug resistance. Pulmonary infections are quite common in HIV patients, and they are the most susceptible group with respect to the empirical usage of antibiotics, especially in low-income countries where access to laboratory diagnosis is limited. The largest burden of fatal antimicrobial resistance corresponds to pulmonary infections observed in the general population of the WHO African region [36].

Trimethoprim-sulfamethoxazole is a broad-spectrum antibiotic combination and has lately been recommended for the prophylaxis of HIV patients in Africa, both symptomatic as well as asymptomatic cases with $CD_4$ T cell count of $\leq 500$ cells/mm$^3$ [36]. Long-term prophylaxis has contributed to a worsening of the extent of bacterial resistance, which spreads within bacterial communities and can cause therapeutic difficulties [34].

In the current study, 51.3% of the bacteria were resistant to trimethoprim-sulfamethoxazole, as noted in some studies done in Ethiopia [13] and Nigeria [26]; however, it was less pronounced than that reported from Dessie, where resistance to this combination of the drug was 69.5% [28]. The frequent and prolonged use of this combination of drug might have contributed to the emergence of resistance. An earlier study done in the same locality reported that 70.5% (n = 24) of nasally colonized *S. pneumoniae* in HIV patients were resistant to the same drug [34]. Higher mortality rates associated with trimethoprim-sulfamethoxazole-resistant infections were reported among the general population of WHO-African regions [36].

Antimicrobial resistance patterns differ geographically and even among hospitals in the same region. Treating acute respiratory tract infections caused by MDR bacteria is exceptionally challenging, and it is linked to increased morbidity and mortality; hence, maintaining local antimicrobial resistance data is important. We found that Gram-positive bacteria were resistant to penicillin (60%), erythromycin (46.3%), and trimethoprim-sulfamethoxazole (39%); however, to a lower extent than that reported by a study formerly conducted at Felege Hiwot Referral Hospital, in the country, in which, Gram-positive bacteria were more resistant to penicillin (68.5%) and the combination drug, trimethoprim-sulfamethoxazole (52.5%) [37].

This study showed that 77.8% of *S. aureus* isolates were resistant to penicillin, 55.6% were resistant to erythromycin, whereas only 37% were resistant to trimethoprim-sulfamethoxazole; alarmingly, 40.7% of them were methicillin-resistant, but is less prominent than that found in a couple of earlier Ethiopian cities, (44.4% and 52.9%) [4,13]. On the other hand, the number of MRSA detected in this work exceeds that found in an Indian study, 19.6% [38]. A former study done in Arba Minch reported that 20.8% of HIV patients had nasal carriage of MRSA [39]. The treatment of MRSA infections poses a significant challenge, particularly in low-income countries where the availability of vancomycin and linezolid are limited, and these drugs create safety issues in immuno-comprised patients; the cost is also prohibitive [40].

The Gram-negative bacteria were resistant to multiple antibiotics, i.e., 58.2%, 71.6%, and 65.3% resistance against trimethoprim-sulfamethoxazole, tetracycline, and amoxicillin-clavulanate, respectively. However, these percentages are lower than that shown by a study reported from Nigeria (amoxicillin-clavulanate (94.1%) and trimethoprim-sulfamethoxazole (75.0%)) [26]. On the other hand, we found that Gram-negative bacteria showed higher degrees of susceptibility to piperacillin (73.1%), cefepime (62.7%), and chloramphenicol (55.2%); the

predominant isolate, i.e., *K. pneumoniae*, was resistant to tetracycline, trimethoprim-sulfamethoxazole, amoxicillin-clavulanate; the extent of resistance being 82%, 71.8%, and 71.8%, respectively. It is important to note that 59% of *K. pneumoniae* isolates were MDR, which is, however, lower than the extent, 93.4% reported recently from Ethiopia [28]. Another important finding of our study is the frightening increase in the incidence of meropenem-resistant Gram-negative bacteria, which stood at 51%. It is noteworthy that the observed rate of severe resistance was unexpected, given the limited utility of this class of antibiotics in the country. Albeit *P. aeruginosa* was a rare cause of typical pneumonia in our study, certain isolates showed resistance to piperacillin, meropenem, and ciprofloxacin and are quite disturbing as the infections caused by these pathogens in HIV patients can be life-threatening.

In general, we have noticed a greater extent of drug resistance among bacterial isolates towards tetracycline, trimethoprim-sulfamethoxazole, ciprofloxacin, and amoxicillin-clavulanate. This could have resulted from the antibiotic therapy and the associated antibiotic pressure, or it can be even due to the inappropriate prescriptions and indiscriminate usage of antibiotics. Also incorrect dosages and schedules for taking antibiotics for a much shorter period would contribute to such an outcome [40].

Our results revealed that 59.6% of isolates were multi-drug resistant, which was lower than that found in a couple of previous studies reported in the country (Bahirdar (77.9%) and Dessie (84.6%)) [13,28]. However, the extent of resistance is much more severe than that reported from another part of Ethiopia,17.9% [4], and Nepal, 52.83% [24]. Isolates of *K. pneumoniae* exhibited higher levels of multi-drug resistance. The growing prevalence of antibiotic resistance among respiratory pathogens may lead to an extension of treatment durations and an enhanced possibility of co-infections. The severe resistance to antibiotics exhibited by bacteria necessitates a reassessment of empirical treatment guidelines, highlighting the importance of proper usage of antibiotics in daily practice within hospital settings.

In this study, 28.1% (31/110) and 20.9% (23/110) of the Gram-negative bacteria were ESBL and carbapenemase producers, respectively, which is considerably lower than the range, 42.1% and 43.4% ESBL reported in a couple of studies in Nepal [24,25]. However, 5.4% (6/110) of the isolates were co-producers of ESBL and carbapenemase. Our results are consistent with the outcome of a recent study conducted in the same locality among HIV patients [41]. In that study, 30% and 16.7% of Gram-negative uro-pathogens were ESBL and carbapenemase producers, respectively. The set of results obtained in the present work indicated that ESBL and carbapenemase production are very much associated with the isolates of *K. pneumoniae*. The WHO Global Report on Antimicrobial Resistance Surveillance has revealed that more than 50% of Klebsiella species are resistant to carbapenem [40]. Hypervirulent pathotypes of *K. pneumoniae*, which recently emerged, possess virulence factors such as hyper-mucoviscous and multiple siderophores, leading to severe debilitating infections [42], and this deserves a special mention in this context.

The current study hints at an escalating trend of MDR, likely stemming from the excessive use of antibiotics and lack of surveillance schemes. Obviously, this is reflected in the case of bacterial pathogens associated with other morbidities existing among HIV patients in the locality [41,43]. The lack of a well-organized bacteriology laboratory and skilled microbiologists exacerbates the problem. Also, poor infection control practices, inadequate hygiene, and self-prescriptions promote the spread of MDR organisms [16]. Implementing preventive measures is pivotal in minimizing the infectivity and safeguarding the health of HIV patients. The outcomes of the current study emphasize the need for establishing a routine local surveillance scheme in the study settings.

The mitigation of typical pneumonia needs a concerted effort from different angles, including ART physicians, nurses, microbiologists, patients, policymakers, and healthcare

administrators. As far as can be ascertained, this is the first bi-centric study related to the magnitude of typical pneumonia in HIV patients in Arba Minch, the results of which have serious implications in connection with the management of typical pneumonia. The magnitude of culture-proven typical pneumonia in the study settings was more severe than anticipated. We have identified some common bacterial pathogens (*K. pneumoniae*, *S. aureus*, and *E. coli*) circulating among the population of HIV patients. Also, this work highlights a paradigm shift in the type of bacteria, i.e., from the predominantly Gram-positive to Gram-negative bacteria, which has important implications associated with the treatment of bacterial infections, as the latter group is more resistant to antibiotics. The WHO-prioritized superbugs such as ESBL, CRE, and MRSA detected in our research may have serious consequences related to the transmission of drug-resistant pathogens, causing the spread of resistance strains among the family members and the community at large. The insights gained from this work can guide clinicians and healthcare professionals in selecting appropriate treatment options, considering the susceptibility patterns. We also envisaged that this study can alert infection prevention and control department to frame new policies to address this issue.

Shortcomings of the present work include a confined cross-sectional study design. A non-probability (consecutive) sampling technique was employed to recruit the study participants. Our study population is comprised of outpatients; this could limit generalizability. In view of the lack of facilities, antisera serotyping was not performed in the case of some of the isolates. In addition, the molecular identification of species and antibiotic-resistant genes of the bacterial isolates was not carried out due to the lack of proper infrastructure.

## Conclusions

This is the first report on the magnitude of typical pneumonia among HIV patients in Arba Minch. The findings of this study have important implications with respect to the current clinical practices and policies. It was found that the overall magnitude of typical pneumonia among HIV patients is on the rise, which highlights the need for frequent sputum analysis and careful diagnosis, followed by subsequent treatment aimed at minimizing the morbidity. Another disappointing fact is the detection of MRSA, ESBL, and CRE-producing strains listed as top-priority pathogens by WHO, revealing that antimicrobial resistance is an impending threat to HIV patients. Accordingly, effective and constant surveillance is warranted in the study area.

## Supporting information

**S1 File. Raw data.**
(XLSX)

## Acknowledgments

The authors thank the College of Medicine and Health Sciences, Arba Minch University, and Arba Minch General Hospital. Thanks are extended to Dr. Sabu KR for English corrections.

## Author Contributions

**Conceptualization:** Ayele Assefa, Melat Woldemariam, Addis Aklilu, Dagninet Alelign, Abdurezak Zakir, Aseer Manilal.

**Data curation:** Ayele Assefa, Melat Woldemariam, Addis Aklilu, Dagninet Alelign, Abdurezak Zakir, Aseer Manilal, Temesgen Mohammed.

**Formal analysis:** Ayele Assefa, Melat Woldemariam, Addis Aklilu, Dagninet Alelign, Abdurezak Zakir, Aseer Manilal, Temesgen Mohammed, Reham M. Alahmadi, Gurusamy Raman, Akbar Idhayadhulla.

**Investigation:** Ayele Assefa, Melat Woldemariam, Addis Aklilu, Dagninet Alelign, Abdurezak Zakir, Aseer Manilal, Reham M. Alahmadi, Gurusamy Raman, Akbar Idhayadhulla.

**Methodology:** Ayele Assefa, Melat Woldemariam, Addis Aklilu, Dagninet Alelign, Abdurezak Zakir, Aseer Manilal, Temesgen Mohammed, Reham M. Alahmadi, Gurusamy Raman, Akbar Idhayadhulla.

**Project administration:** Ayele Assefa.

**Resources:** Ayele Assefa.

**Software:** Ayele Assefa, Melat Woldemariam, Dagninet Alelign, Abdurezak Zakir, Aseer Manilal, Temesgen Mohammed.

**Supervision:** Melat Woldemariam, Addis Aklilu, Dagninet Alelign, Abdurezak Zakir, Aseer Manilal, Temesgen Mohammed, Reham M. Alahmadi, Gurusamy Raman, Akbar Idhayadhulla.

**Validation:** Ayele Assefa, Melat Woldemariam, Addis Aklilu, Dagninet Alelign, Abdurezak Zakir, Aseer Manilal, Temesgen Mohammed, Reham M. Alahmadi, Gurusamy Raman, Akbar Idhayadhulla.

**Visualization:** Ayele Assefa, Melat Woldemariam, Addis Aklilu, Dagninet Alelign, Abdurezak Zakir, Aseer Manilal, Temesgen Mohammed.

**Writing – original draft:** Ayele Assefa, Melat Woldemariam, Addis Aklilu, Aseer Manilal, Reham M. Alahmadi, Gurusamy Raman, Akbar Idhayadhulla.

**Writing – review & editing:** Ayele Assefa, Melat Woldemariam, Addis Aklilu, Dagninet Alelign, Abdurezak Zakir, Aseer Manilal, Temesgen Mohammed, Reham M. Alahmadi, Gurusamy Raman, Akbar Idhayadhulla.

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
