## [Decision Letter · Decision Letter 0]

29 Jan 2024

PONE-D-23-41432Typical Pneumonia among HIV infected Patients in Public Hospitals, southern Ethiopia.PLOS ONE

Dear Dr. Aklilu,

Thank you for submitting your manuscript to PLOS ONE. After careful consideration, we feel that it has merit but does not fully meet PLOS ONE’s publication criteria as it currently stands. Therefore, we invite you to submit a revised version of the manuscript that addresses the points raised during the review process.

We look forward to receiving your revised manuscript.

Kind regards,

Mengistu Hailemariam Zenebe, PhD

Academic Editor

PLOS ONE

Journal Requirements:

2. Please include a separate caption for each figure in your manuscript.

Additional Editor Comments:

Dear Author,

After analyzing your reviewers' comments, I realized that creating the best version for publication was essential. Could you, therefore, make the necessary modifications to the helpful feedback provided?

Best

Reviewers' comments:

Reviewer's Responses to Questions

**Comments to the Author**

1. Is the manuscript technically sound, and do the data support the conclusions?

Reviewer #1: Yes

Reviewer #2: Partly

Reviewer #3: Partly

2. Has the statistical analysis been performed appropriately and rigorously? 

Reviewer #1: I Don't Know

Reviewer #2: Yes

Reviewer #3: Yes

3. Have the authors made all data underlying the findings in their manuscript fully available?

Reviewer #1: Yes

Reviewer #2: Yes

Reviewer #3: No

4. Is the manuscript presented in an intelligible fashion and written in standard English?

Reviewer #1: Yes

Reviewer #2: No

Reviewer #3: Yes

5. Review Comments to the Author

Reviewer #1: Review report (PLOS ONE)

Date: 21 January, 2024

The manuscript entitled ‘Typical Pneumonia among HIV infected patients in public Hospitals, Southern Ethiopia’ by Aklilu et al attempts to inform the prevalence of drug resistant bacteria causing pneumonia among HIV infected subjects in the given study area. I am impressed by the ample sample size, coherence in writing and ideas, and the compelling findings. I therefore support this article to appear in the ‘Plos One’. However, the authors may want to improve on the following sections unless the article is not acceptable in the present form.

Comments

Introduction

1. This section sounds good which has been able to articulate ideas. I suggest authors add a or two sentence(s) on the importance of pyogenic bacteria among HIV-related infections.

2. L20-23: The sentence is not clear. I suggest authors rewrite the sentence.

Methods

1. There seem to be some structural differences among the two types of hospitals: general and primary. How are the HIV-related cases managed differently in these hospitals? I suggest authors describe it in a or two sentence(s).

2. I suggest authors provide figures of the subjects on each of the excluded criteria and total approached subjects.

3. Include methods (phenotypic and confirmatory) describing ESBL and carbapenem resistant tests.

Discussion

1. I suggest authors add a small paragraph on the strengths, limitations, and clinical as well as policy implications of the study.

Conclusion

1. This section needs a major revision. I suggest authors trim the description and rewrite the more meaningful conclusion.

L 434: Citation format needs to be adjusted.

Table 4: Write the full form of all abbreviations.

Above all, the article needs further improvement in standard English and grammatical errors.

Reviewer #2: Thank you for inviting me to review the manuscript. Although the research idea is very interesting, it lacks some clarity. The authors should consider the following issues:

1. There is no clear information about the type of pneumonia (Community-acquired vs hospital-acquired).

2. There are grammar mistakes and redundancy of words (example: the word “In fact” in lines 8, 9, and 17, “Ipso facto” in line 41, “area, setting” in line 62, “opted” in line 77….etc).

3. Abstract: the sentence “Sputum were collected and inspected for pyogenic bacteria following standard…..”. Why you mention pyogenic here? In my point of view, it is misunderstanding.

4. Line 99: “Expectorated specimens (3-5ml) were collected in a sterile sputum cup (falcon tube)”. How you collect sputum in HIV patients that are unable to produce and give sputum specimen? Have you used sputum inducer? Again, sputum cup and falcon tube are different materials.

5. Line 104: Procedure for bacterial identification should be written in detail.

6. Line 107-108: Blood agar plate should be incubated in 5% carbon dioxide generating candle jar at 37˚C for 24 hours

7. Line 214: Define MDR

Reviewer #3: This work is very important and interesting in trying to assess prevalence of typical pneumonia among vulnerable population.

However, there are major revisions required to the writing style of the manuscript, especially abstract and methods.

I have suggestion on the following issue.

Abstract part

-it says “A pretested semi-structured questionnaire was employed to collect the demographic” what does it mean semi-structured, please correct as structured.

- It says, “A total of 152 pyogenic bacteria were identified, comprising 73% (n=111) Gram negative”. Gram negatives what? Please correct Gram negative as Gram negative isolate or Gram-negative bacteria.

- It says “Two-score (11/27) of S. aureus was MRSA, and 26.6 (n=29) and 21.1% (n=23) of Gram-negative isolates were ESBL and carbapenemase producers, respectively”. What is Two-score? And please correct the percentage of ESBL like this 26.6%.

- All abbreviations, such as MRSA should be defined on first use, generally abbreviations is not recommended in abstract part.

-please correct Keywords as follows: typical pneumonia, bacterial isolates, pyogenic, antimicrobial susceptibility pattern, HIV, Arba Minch, Ethiopia.

Introduction part

-line 8-9: the author uses two times in fact, it indicated that there is self-plagiarism, so, please correct it.

-line 10 and 21: AIDS and WHO should be used long form with abbreviations

Methods and Materials

- There are like AMGH and DFPH please use long form.

-for Bacterial isolation method are like conventional biochemical method and molecular methods are used, so which method was used for identification of bacterial species like K. pneumoniae, S. aureus E. coli, S. pyogenes, P. aeruginosa, E. aerogenes, and S. pneumoniae. Please describe all method you have used for identification for all bacterial isolates.

-How do you identify MRSA

-the authors use sputum sample but there is not mycobacterium tuberculosis in the result part, because this bacterium one of the causative agents of pneumonia, if so, please say something in the method part or inclusion and exclusion criteria.

- For culture media the author must write manufactured company, and the country

- The authors should be mentioned all antibiotics with their concentration, manufactured company, and the country,

-line 135: ATCC should be used long form with abbreviations, then you can use abbreviations

- Ethical considerations. What was the fate of those patients positive for pyogenic bacteria?

Result part

- “Ok” but please increase the resolution of figures

-The discussion is OK. But it needs some improvements.

- Please see again the conclusions; it is not based

6. PLOS authors have the option to publish the peer review history of their article (what does this mean?). If published, this will include your full peer review and any attached files.

Reviewer #1: No

Reviewer #2: **Yes: **Muluneh Assefa

Reviewer #3: No

---

## [Author Response · Author response to Decision Letter 0]

9 Mar 2024

We have corrected the comments point by point raised by reviewers and we have attached the response to reviewers in the uploaded files. Therefore, please refer the response to reviewers for detailed information.

---

## [Decision Letter · Decision Letter 1]

13 May 2024

PONE-D-23-41432R1Typical Pneumonia among Human Immunodeficiency Virus-infected Patients in Public Hospitals in Southern Ethiopia. Dear Dr. Aklilu,I thank you for the amendment but you need to fix more on the given comments.

Thank you for submitting your manuscript to PLOS ONE. After careful consideration, we feel that it has merit but does not fully meet PLOS ONE’s publication criteria as it currently stands. Therefore, we invite you to submit a revised version of the manuscript that addresses the points raised during the review process.

We look forward to receiving your revised manuscript.

Kind regards,

Mengistu Hailemariam Zenebe, PhD

Academic Editor

PLOS ONE

Journal Requirements:

Reviewers' comments:

Reviewer's Responses to Questions

**Comments to the Author**

1. If the authors have adequately addressed your comments raised in a previous round of review and you feel that this manuscript is now acceptable for publication, you may indicate that here to bypass the “Comments to the Author” section, enter your conflict of interest statement in the “Confidential to Editor” section, and submit your "Accept" recommendation.

Reviewer #2: (No Response)

Reviewer #4: (No Response)

2. Is the manuscript technically sound, and do the data support the conclusions?

Reviewer #2: Yes

Reviewer #4: Yes

3. Has the statistical analysis been performed appropriately and rigorously? 

Reviewer #2: Yes

Reviewer #4: Yes

4. Have the authors made all data underlying the findings in their manuscript fully available?

Reviewer #2: Yes

Reviewer #4: Yes

5. Is the manuscript presented in an intelligible fashion and written in standard English?

Reviewer #2: Yes

Reviewer #4: Yes

6. Review Comments to the Author

Reviewer #2: Dear Authors,

I appreciate your feedback on my suggestions. But you should consider my previous comment: "Blood agar plates should be incubated in a 5-10% carbon dioxide generating candle jar at 37˚C for 24 hours like a chocolate agar plate. Again, in your manuscript, you have used the word "pyogenic" repeatedly. I'm not satisfied with the way you mentioned the bacterial species that cause productive coughs. I think you should differentiate pyogenic bacteria, which means pus-forming bacteria such as S. aureus and S. pyogenes. Sputum is expectorated from the lungs and composed of mucus that may contain pus (dead WBCs), blood, fibrin, or bacteria. I advised you to remove this word. Thank you!

Reviewer #4: Abstract

-Methods section: What type of culture media and biochemical tests used for bacterial identification?

-Conclusion section: what do you recommend?

Materials and Methods

-Do you considered local availability of antibiotics for AST while you choose antibiotics in addition to following CLSI guidelines?

Discussion

-What factors attribute for shift in prevalence from gram positive to gram negative bacteria?

-Previously S. aureus is most prevalent in typical pneumonia, in your study K. pnuemoniae was most prevalent, So how do you explain this?

7. PLOS authors have the option to publish the peer review history of their article (what does this mean?). If published, this will include your full peer review and any attached files.

Reviewer #2: No

Reviewer #4: **Yes: **Dr. Belayneh Regasa Dadi

---

## [Author Response · Author response to Decision Letter 1]

15 May 2024

We have carefully addressed all the comments and queries raised by the editor and reviewers. We have attached a separate file of response to reviewers so please refer for details.

---

## [Editor Report · Decision Letter 2]

20 May 2024

PONE-D-23-41432R2Typical Pneumonia among Human Immunodeficiency Virus-infected Patients in Public Hospitals in Southern Ethiopia.PLOS ONE

Dear Dr. Aklilu,

Thank you for submitting your manuscript to PLOS ONE. After careful consideration, we feel that it has merit but does not fully meet PLOS ONE’s publication criteria as it currently stands. Therefore, we invite you to submit a revised version of the manuscript that addresses the points raised during the review process.

We look forward to receiving your revised manuscript.

Kind regards,

Mengistu Hailemariam Zenebe, PhD

Academic Editor

PLOS ONE

Journal Requirements:

**Additional Editor Comments:**

Dear Author,

I haven't get your document that you say separate file response for reviewer

would you incorporate please?

---

## [Author Response · Author response to Decision Letter 2]

22 May 2024

We have considered and corrected all comments given by the editor and peer reviewers carefully. We have attached the separate file of response to reviewers. Therefore, you can get the detailed response from this file.

---

## [Editor Report · Decision Letter 3]

11 Jul 2024

Typical Pneumonia among Human Immunodeficiency Virus-infected Patients in Public Hospitals in Southern Ethiopia.

We’re pleased to inform you that your manuscript has been judged scientifically suitable for publication and will be formally accepted for publication once it meets all outstanding technical requirements.

Kind regards,

Mengistu Hailemariam Zenebe, PhD

Academic Editor

PLOS ONE
---

## [Editor Report · Acceptance letter]

22 Jul 2024

PONE-D-23-41432R3 

PLOS ONE

Dear Dr. Aklilu, 

I'm pleased to inform you that your manuscript has been deemed suitable for publication in PLOS ONE. Congratulations! Your manuscript is now being handed over to our production team.

Kind regards, 

on behalf of

Dr. Mengistu Hailemariam Zenebe 

Academic Editor

PLOS ONE